# OmpK36-mediated Carbapenem resistance attenuates ST258 *Klebsiella pneumoniae* in vivo

Joshua L.C. Wong[1,2,3], Maria Romano[3,4], Louise E. Kerry [1,3], Hok-Sau Kwong [3,4], Wen-Wen Low[1,3], Stephen J. Brett [2], Abigail Clements[1,3], Konstantinos Beis [3,4] & Gad Frankel [1,3]

Carbapenem-resistance in *Klebsiella pneumoniae* (KP) sequence type ST258 is mediated by carbapenemases (e.g. KPC-2) and loss or modification of the major non-selective porins OmpK35 and OmpK36. However, the mechanism underpinning OmpK36-mediated resistance and consequences of these changes on pathogenicity remain unknown. By solving the crystal structure of a clinical ST258 OmpK36 variant we provide direct structural evidence of pore constriction, mediated by a di-amino acid (Gly115-Asp116) insertion into loop 3, restricting diffusion of both nutrients (e.g. lactose) and Carbapenems. In the presence of KPC-2 this results in a 16-fold increase in MIC to Meropenem. Additionally, the Gly-Asp insertion impairs bacterial growth in lactose-containing medium and confers a significant in vivo fitness cost in a murine model of ventilator-associated pneumonia. Our data suggests that the continuous selective pressure imposed by widespread Carbapenem utilisation in hospital settings drives the expansion of KP expressing Gly-Asp insertion mutants, despite an associated fitness cost.

[1] Centre for Molecular Bacteriology and Infection, Department of Life Sciences, Imperial College London, London, UK. [2] Department of Surgery and Cancer, Section of Anaesthetics, Pain Medicine and Intensive Care, Imperial College London, London, UK. [3] Department of Life Sciences, Imperial College London, London, UK. [4] Rutherford Appleton Laboratory, Research Complex at Harwell, Didcot, Oxfordshire, UK. Correspondence and requests for materials should be addressed to G.F. (email: g.frankel@imperial.ac.uk)

The acquired resistance to Carbapenems by *Klebsiella pneumoniae* (KP) and other Gram-negative organisms is an increasing global problem that potentially jeopardises the future utility of a fundamentally important antibiotic class used for the treatment of life-threatening infections[1]. KP infection is usually hospital-acquired where it accounts for around 30% of Gram-negative infections[2]. In a US study covering long-term facilities between 2014 and 2015, almost 25% of KP isolates were Carbapenem-resistant (CRKP)[3]. CRKP is now endemic in some regions[4] and are classified as 'critical' WHO Priority 1 organisms[5]. Hospital-acquired CRKP infection mortality is high and there may be an increased risk of death when infection is caused by resistant versus sensitive KP strains (42 vs 21%)[6]. Importantly, while this may be due to the potential inferiority and known toxicity of the few alternative antibiotics, it illustrates the success of KP as a key antibiotic resistant pathogen and our reliance on Carbapenems.

The expression of carbapenemase enzymes, usually encoded on large resistance plasmids[7], represents the first genetic source of resistance. These enzymes inactivate Carbapenems by hydrolysis. The plasmids are transferred vertically from parental to daughter cells during cell division or horizontally by conjugal transfer. Chromosomally, modification of the major outer membrane porins, OmpK35 and OmpK36, limit antibiotic influx across the outer membrane in CRKP[8]. These changes act in concert to effectively lower active Carbapenem concentrations at the site of their transpeptidase targets, the periplasm, abrogating their bactericidal effect.

One multilocus sequence type (MLST), ST258, has been internationally successful in driving CRKP dissemination[9]. ST258 strains have closely expanded with the KPC family of carbapenemases[4], which in tandem with modifications in OmpK35 and OmpK36, afford clinically relevant (high) minimum inhibitory concentrations (MICs) to Carbapenems. However, porins play important physiological roles and facilitate both the influx of small hydrophilic solutes, including nutrients, and

the efflux of toxic products across the otherwise impermeable Gram-negative outer membrane[10]. In keeping with this, deletion of both porin genes results in attenuation during in vivo infection[11]. The overall structures of OmpK35 and OmpK36 display similar features to other general porins, a trimeric architecture composed of 16-stranded β-barrels[12,13]. Two important structural components in porins are extracellular loops 3 and 4 (L3 and L4). L3 is not exposed at the cell surface but folds back into the barrel, forming a constriction zone half way inside the channel that contributes to the permeability properties, such as size exclusion limit and ion selectivity of the pore[12]. L4 lies away from the pore and is involved in monomer trimerisation and subsequent stability.

OmpK35 is ubiquitously truncated in ST258 strain collections[14], where a common mutation (Genbank FJ577672) encodes a frame-shift that results in a premature stop codon (TGA) and large truncation (Fig. 1a, Supplementary Fig. 1a). This mutation results in an unstructured and non-functional pore due to the encoding of only a small conserved 19 amino acid N-terminal fragment following signal peptide cleavage. There is more heterogeneity in OmpK36 sequences[14]. We chose to study an OmpK36$_{ST258}$ variant from an ST258 strain (KPST258, Supplementary Table 1) that exhibits high Carbapenems MICs, which is associated with increased mortality[6]. The OmpK36$_{ST258}$ protein sequence of KPST258 is 100% identical to the consensus accession WP_00415112 present in 1132 NCBI assemblies. In comparison with the reference laboratory strain ATCC43816, the OmpK36 variant includes a Gly115-Asp116 (GD) insertion after the conserved Pro109-Glu-Phe-Gly-Gly-Asp114 motif in L3 and a Leu165-Ser-Pro167 (LSP) insertion in L4 (Fig. 1b, Supplementary Fig. 1b). The GD insertion and other sequence variations in L3 of OmpK36$_{ST258}$ are correlated with increased resistance in clinical isolates from international KP collections[15–17].

KP ST258 is evidently flourishing and a reduced permeability barrier is clearly beneficial in the face of prevailing antibiotic selection. This prompted us to explore the precise molecular

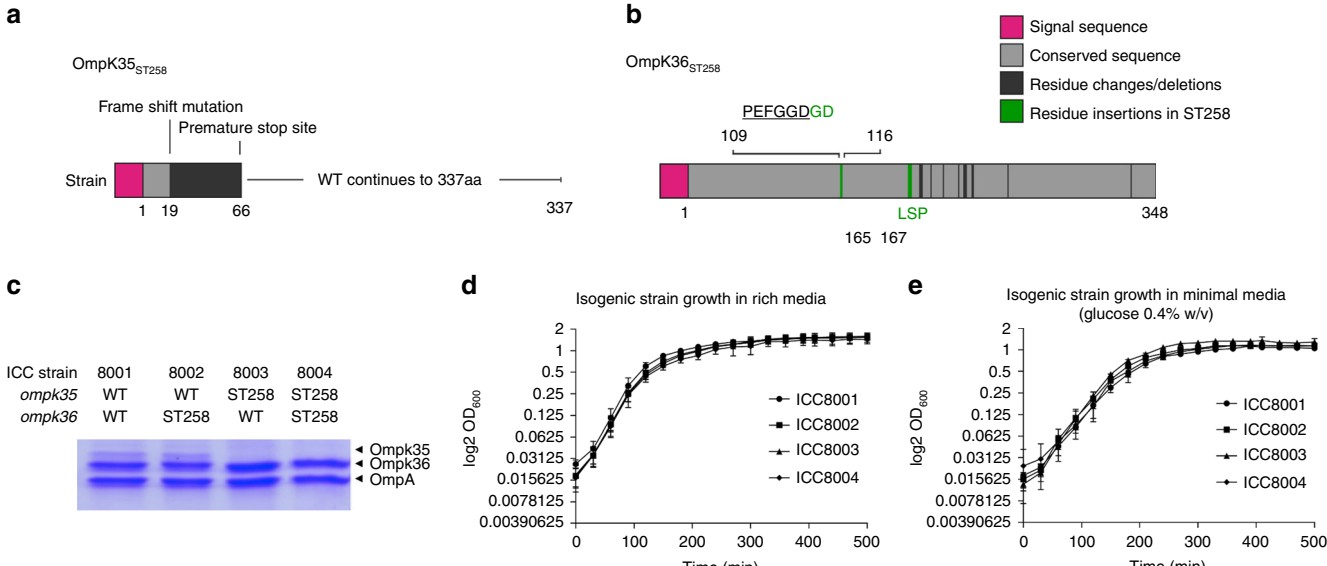

**Fig. 1** OmpK35$_{ST258}$ and OmpK36$_{ST258}$ porin variants do not impact on growth in vitro. **a** The OmpK35$_{ST258}$ frame-shift mutation results in a premature TGA stop codon. Sequence conservation with OmpK35$_{WT}$ is lost within the first β-strand resulting in a truncated protein. **b** OmpK36 sequences share 95% identity, with two insertions (GD in loop 3 and LSP in loop 4) in OmpK36$_{ST258}$ (highlighted in green). The underlined PEFGGD is a conserved loop 3 motif in OmpK36 porins. **c** Outer membrane preparations demonstrate a loss of OmpK35 in ICC8003 and ICC8004. OmpK36$_{WT}$ (ICC8001 and ICC8003) and OmpK36$_{ST258}$ (ICC8002 and ICC8004) are present in similar abundance in isogenic strains. **d**, **e** Growth, measured by OD$_{600}$, is not affected by the introduction of ST258 porins into ICC8001 in rich (Luria Bertani) media (**d**) or minimal (M9) media (**e**) with glucose as the sole carbon source ($n = 3$ repeats, error bars = s.d.)

resistance mechanism underlying changes in the outer membrane and to determine if these come at a fitness cost. Here we show that the GD insertion in OmpK36$_{ST258}$ constricts the pore and restricts diffusion of Carbapenems and the disaccharide lactose across membranes. Pore constriction attenuates KP growth in medium containing lactose as a sole carbon source in vitro and infection in a murine model of ventilator-associated pneumonia in vivo. These results suggest that the selective pressure posed by the extensive usage of Carbapenems is a major driver in the continuous spread of CRKP, a key pathogen with a significant attributable morbidity, mortality and socioeconomic cost[18].

## Results and Discussion

**ST258 outer membrane porins enhance Carbapenem resistance.** We investigated the impact of the sequence variations found in OmpK35 and OmpK36 of KPST258 on Carbapenem resistance. To this end, we substituted the endogenous porin genes in a Rif$^R$ derivative of ATCC43816, we named ICC8001 (Supplementary Table 1), with OmpK36$_{ST258}$ (ICC8002), OmpK35$_{ST258}$ (ICC8003) or both (ICC8004) (Fig. 1c and Table 1). The absence of OmpK35 in ICC8003 and ICC8004, resulting from the introduction of the ST258 truncation mutant (66aa vs 337aa WT sequence), was confirmed by analysis of outer membrane preparations (Fig. 1c). Replacement of the OmpK36 coding sequence alone, between the endogenous chromosomal promoter and terminator regions at the wild-type (WT) locus of ICC8001, resulted in a similar abundance of OmpK36$_{WT}$ and Ompk36$_{ST258}$ in all four strains (Fig. 1c). No growth defects were detected when the different strains were grown in vitro in either rich (Luria Bertani, LB) or minimal (M9) media containing glucose (0.4% w/v) as the sole carbon source (Fig. 1d, e). This suggested that the ST258 porin variants do not adversely affect KP's ability to grow in extremes of extracellular osmotic pressure or when glucose is provided as the sole carbohydrate available for metabolism.

We next assessed the impact of sequential ST258 porin gene substitution on antimicrobial resistance in the absence or presence of the carbapenemase genes KPC-2 and OXA-48, encoded on epidemic pKpQIL-like and pOXA-48a-like circulating plasmids. Utilising a reference laboratory broth MIC panel, designed to evaluate resistant Gram-negative organisms, revealed that in the absence of carbapenemases all the strains remain Carbapenem sensitive (Supplementary Fig. 2a). Moreover, the isogenic strains remain sensitive to aminoglycosides and Tigecycline where porin loss or mutation do not influence susceptibility (Fig. 2e).

Both OmpK35$_{ST258}$ truncation (ICC8003) and OmpK36$_{ST258}$ substitution (ICC8002) increased resistance to Carbapenems in the presence of KPC-2 and OXA-48 (Fig. 2a, b, c, d), although the absolute MIC values for these agents were found to be dependent on the enzymatic activity of each carbapenemase. The lower levels of resistance achieved by OXA-48 (Ambler class D) expressing strains is attributable to a weaker hydrolytic activity towards

Carbapenems than that mediated by KPC-2 (Ambler class A)[19]. Of note, the contribution of OmpK36$_{ST258}$ to Carbapenem resistance was greater than OmpK35$_{ST258}$. For example, the MIC to Meropenem (Fig. 2a, d) is 1 mg/L in ICC8001, 8 mg/L in ICC8003 (encoding OmpK35$_{ST258}$) and 16 mg/L in ICC8002 (encoding OmpK36$_{ST258}$) in strains expressing KPC-2. This demonstrates that the WT OmpK35 and OmpK36 porins in ICC8001 allow sufficient periplasmic diffusion and bactericidal activity for this strain to remain below the Meropenem sensitivity breakpoint (2 mg/l), despite KPC-2 and OXA-48 mediated hydrolysis. This pattern extends to other classes of drugs with similar mechanisms of action, such as the third and fourth generation Cephalosporins, Cefotaxime and Cefepime (Fig. 2e). The sensitivity to Ceftazidime is restored by the novel diazabicyclooctane (DBO) non-β-lactam β-lactamase inhibitor, Avibactam (Fig. 2e), whereas other OmpK36 variants have been shown to contribute to resistance against this new agent[20]. Importantly, the MIC to Meropenem of ICC8004 (encoding OmpK35$_{ST258}$ and OmpK36$_{ST258}$) was 32 mg/L, which reproduced the Carbapenem antibiogram (together with resistance to Imipenem and Ertapenem) of the KPST258 strain harbouring the pKpQIL-like plasmid (Fig. 2d). ICC8004 far exceeded the Carbapenem resistance breakpoints of the agents tested. Indeed, the antibiotic levels required for treating KP, expressing this ST258 porin configuration, would be unachievable even in the context of attempting to optimise dosing by continuous infusion in current trials (NCT03213990).

**OmpK36 pore constriction underpins Carbapenem resistance.** The robust increase in Carbapenem resistance conferred by the expression of OmpK36$_{ST258}$ compared with OmpK36$_{WT}$, despite both proteins sharing 95% identity, prompted us to elucidate the underlying molecular resistance mechanism. As the OmpK36 used in previous structural work contains a Q235R mutation (PDB ID: 5O79), which is found behind L3 and therefore could have affected its conformation, we started by solving the structure of the WT OmpK36 (PDB ID: 6RD3, Table 2). This revealed that the two structures can be superimposed with an rmsd of 0.18 Å over 340 C$_\alpha$ atoms, suggesting that the Q235R substitution had no effect on the conformation of L3. We next solved the structure of OmpK36$_{ST258}$ at 3.23 Å resolution by molecular replacement, using both OmpK36$_{WT}$ and the OmpK36$_{Q235R}$ as search models[13] (Fig. 3a, Table 2). The overall architecture of the OmpK36$_{ST258}$ porin is preserved despite multiple sequence variations, including L3 and L4 insertions. The OmpK36$_{ST258}$ structure can be superimposed on OmpK36$_{WT}$ with an rmsd of 0.43 Å over 340 C$_\alpha$ atoms. The structure revealed that the GD insertion in L3 of OmpK36$_{ST258}$ resulted in an extended loop conformation that intrudes into the pore, at the constriction zone, which reduced the pore diameter by 26% (3.2 Å WT and 2.37 Å ST258 diameter) (Fig. 3b, Supplementary Fig. 3a). The structure suggests that L3 is further stabilised by the formation of a salt-bridge between D114 and R127 at the barrel face of the pore

**Table 1 Isogenic strains used in this study with their corresponding OmpK35 and OmpK36 composition**

| Strain | OmpK35 | OmpK36 |
|---|---|---|
| ICC8001 | OmpK35$_{WT}$ | OmpK36$_{WT}$ |
| ICC8002 | OmpK35$_{WT}$ | OmpK36$_{ST258}$ |
| ICC8003 | OmpK35$_{WST258}$ | OmpK36$_{WT}$ |
| ICC8004 | OmpK35$_{ST258}$ | OmpK36$_{ST258}$ |
| WT + GD | OmpK35$_{ST258}$ | OmpK36$_{WT+GD}$ with loop 3 Gly115Asp116 insertion |
| ST258ΔGD | OmpK35$_{ST258}$ | OmpK36$_{ST258ΔGD}$ with loop 3 Gly115Asp116 deletion |
| ST258ΔLSP | OmpK35$_{ST258}$ | OmpK36$_{ST258ΔLSP}$ with loop 4 Leu165-Ser-Pro167 deletion |
| ST258R127A | OmpK35$_{ST258}$ | OmpK36$_{ST258R127A}$ with Arg127 to Ala127 mutation |

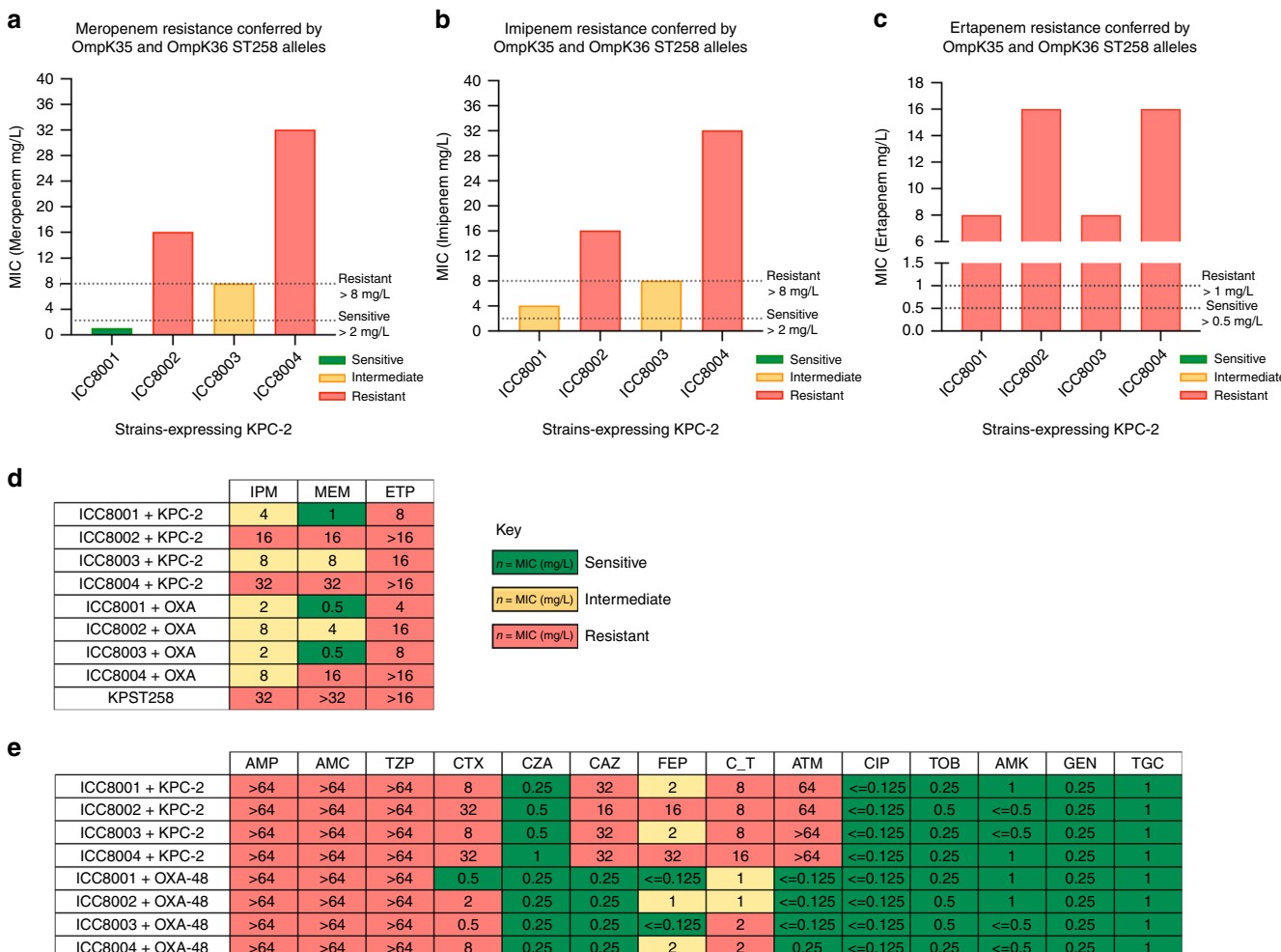

**Fig. 2** The impact of OmpK35$_{ST258}$ and OmpK36$_{ST258}$ substitution on resistance to antibiotics used in Gram-negative infections. **a** Meropenem, **b** Imipenem and **c** Ertapenem broth minimum inhibitory concentrations presented graphically in isogenic strains expressing the KPC-2 carbapenemase. Dotted lines represent sensitive and resistant EUCAST breakpoints. **d** Broth MIC of the isogenic KP strains expressing the KPC-2 or OXA-48 carbapenemase. Individual values are colour coded (green—sensitive, yellow—intermediate and red—resistant) according to their antibiotic resistance defined by EUCAST breakpoints. Antibiotic key: IPM Imipenem, MEM Meropenem, ETP Ertapenem. **e** Resistance to other antibiotics in different classes tested. Individual values are colour coded (green—sensitive, yellow—intermediate and red—resistant) according to their antibiotic resistance defined by EUCAST breakpoints. Antibiotic key: AMP Ampicillin, AMC Amoxicillin/Clavulanate (2:1), TZP Pipericillin/Tazobactam, CTX Cefotaxime, CZA Ceftazidime/Avibactam, CAZ Ceftazidime, FEP Cefipime, C_T Ceftolozone/Tazobactam, ATM Aztreonam, CIP Ciprofloxicin, TOB Tobramycin, AMK Amikacin, GEN Gentamicin, TGC Tigecycline

(Fig. 3a). In order to evaluate whether the L3 GD motif is the sole molecular mechanism contributing to the resistant phenotype we constructed two further mutants for evaluation in subsequent structural and functional experiments. We created an OmpK36$_{WT}$ chimera in which we inserted a GD in L3 (OmpK36$_{WT+GD}$) and a GD deletion mutant where the two amino acid motif was removed from L3 of the resistant OmpK36$_{ST258}$ (OmpK36$_{ST258\Delta GD}$). Solving the OmpK36$_{WT+GD}$ structure at 2.03 Å resolution revealed a similar conformation to OmpK36$_{ST258}$ with formation of a D114/R127 salt-bridge and pore constriction to 2.87 Å (Fig. 3b, Table 2).

To determine if the observed pore diameter reduction, mediated by L3 GD insertion, influenced permeability, we performed liposomal swelling assays and quantified carbohydrate and Carbapenem diffusion across the OmpK36 isoforms. While not diffusing into empty proteoliposomes, glucose (180 g/mol), the smallest carbohydrate tested, freely diffused across all OmpK36 isoforms, demonstrating formation of functional pores (Supplementary Fig. 4b). The L3 GD mediated pore constriction

did not impact on glucose diffusion, in keeping with the in vitro growth curves in minimal media containing glucose as the sole carbon source (Fig. 1e). Stachyose, a tetrasaccharide, with the highest molar mass tested at 666 g/mol was unable to diffuse across either OmpK36$_{WT}$ or OmpK36$_{WT+GD}$ isoforms (Supplementary Fig. 5b). Based on this data we next tested diffusion of the disaccharide lactose (342 g/mol), with a molar mass between that of glucose and stachyose. Diffusion of lactose was impaired in the presence of L3 GD in OmpK36$_{WT+GD}$ compared with OmpK36$_{WT}$ (Supplementary Fig. 5e). Consistent with this finding, an isogenic strain expressing OmpK36$_{WT+GD}$, in M9 containing lactose as the sole carbon source, demonstrated a growth defect compared with the OmpK36$_{WT}$ expressing strain (Supplementary Fig. 5f).

Meropenem (383 g/mol) diffusion was significantly reduced in OmpK36$_{ST258}$-containing proteoliposomes compared with OmpK36$_{WT}$ (Fig. 3c, d). This diffusion barrier is reproduced by GD insertion in OmpK36$_{WT+GD}$, and completely reversed by L3 GD deletion in OmpK36$_{ST258\Delta GD}$ (Fig. 3c, d). Expression of the

**Table 2 Data collection and refinement statistics**

|  | OmpK36$_{ST258}$ | OmpK36$_{WT+GD}$ | OmpK36$_{WT}$ |
|---|---|---|---|
| *Data collection* |  |  |  |
| Space group | P 2$_1$ 2$_1$ 2 | C 1 2 1 | P 1 2$_1$ 1 |
| Cell dimensions |  |  |  |
| *a, b, c* (Å) | 286.46, 326.10, 164.19 | 232.46, 74.41, 90.11 | 55.09, 316.05, 73.79 |
| α, β, γ (°) | 90, 90, 90 | 90, 111.73, 90 | 90, 102.86, 90 |
| Resolution (Å)[a] | 286.03–3.23 (3.57–3.23) | 57.37–2.03 (2.06–2.03) | 53.20–1.98 (2.01–1.98) |
| $R_{sym}$ or $R_{merge}$ | 19.9 (111) | 14.7 (81.1) | 23.9 (134.0) |
| $I/\sigma I$ | 6.6 (1.6) | 4.2 (1.2) | 6.3 (1.2) |
| Completeness (%) spherical | 53.9 (11.9) | 99.8 (98.3) | 100.0 (100.0) |
| Completeness (%) ellipsoidal | 86.4 (60.3) | – | – |
| Redundancy | 4.6 (5.8) | 3.2 (3.1) | 6.9 (6.8) |
| CC(1/2) | 0.99 (0.55) | 0.98 (0.58) | 0.99 (0.50) |
| *Refinement* |  |  |  |
| Resolution (Å) | 91.51–3.27 | 57.73–2.03 | 53.20–1.98 |
| No. of reflections | 127092 | 92323 | 169757 |
| $R_{work}/R_{free}$ | 21.8/24.9 | 20.8/24.6 | 19.4/23.4 |
| No. atoms | 48852 | 8580 | 17283 |
| Protein | 48852 | 8091 | 16114 |
| Ligand/ion | – | 147 | 287 |
| Water | – | 342 | 882 |
| *B-factors* |  |  |  |
| Protein | 76.3 | 29.7 | 24.5 |
| Ligand/ion | – | 48.1 (lipid), 28.9 (LDAO), 41.2 (C$_8$E$_4$) | 36.2 (C$_8$E$_4$) |
| Water | – | 32.1 | 28.1 |
| R.m.s. deviations |  |  |  |
| Bond lengths (Å) | 0.002 | 0.007 | 0.006 |
| Bond angles (°) | 0.64 | 0.85 | 0.83 |

Values in parentheses are for highest-resolution shell
"Spherical" looks at all data within a specific, spherical resolution range/bin, so this is the usual, well-known way of looking at data as a function of resolution. "Ellipsoidal" additionally requires that a data point be within the fitted ellipsoid in order to be considered
[a]All data sets were collected from one crystal

OmpK36 chimeras from the KP genome (in combination with the OmpK35$_{ST258}$ truncation and KPC-2 expression) recapitulated this effect as the MICs to Meropenem are reflective of the presence or absence of the GD insertion in L3 (Fig. 3e). Diffusion and resistance assessed with these methodologies were similar for both Ertapenem and Imipenem (Supplementary Fig. 4). These results suggest that L4 insertions do not play a role in pore reduction, in unison with the structure, where the LSP insertion results in an extended loop conformation that perturbs the top of the barrel (away from the constriction zone). We confirmed this by deleting the L4 LSP motif in OmpK36$_{ST258ΔLSP}$; removal of these residues did not influence resistance to Carbapenems in the presence of KPC-2 (Supplementary Fig. 6). Whilst the calculated pore diameter of the clinical OmpK36$_{ST258}$ (2.37 Å) is smaller than the OmpK36$_{WT+GD}$ chimera (2.87 Å), the data suggest that for these important antibiotics small pore reductions are sufficient to significantly retard influx. In order to assess the contribution of the salt-bridge, we generated an R127A substitution (OmpK36$_{ST258R127A}$). We observed no change in Carbapenem diffusion between this isoform and OmpK36$_{ST258}$ in swelling assays or change in MIC to Carbapenems (Supplementary Fig. 7). This supports a model in which pore constriction, resulting from the L3 GD insertion alone is sufficient to mediate an increase in resistance to Carbapenems in OmpK36$_{ST258}$.

Elucidating the mechanism underlying the limitation of diffusion in OmpK$_{ST258}$ provides important physical parameters that should be considered in future efforts in rationale antimicrobial drug design. For example, we show the minimal pore radius of OmpK$_{ST258}$ is 2.37 Å and this should be taken into account during the physico-chemical drug design stages. Isogenic strains, such as these with contemporary porin substitutions,

could be used to screen new agents to ensure that drug delivery to the pharmacological site of action is not impaired, especially in important commonly antimicrobial resistant pathogens such as the ESKAPE group[1].

**ST258 porins impact on fitness in a severe pneumonia model.** Our results show that the pore constriction directly contributes to Carbapenem resistance and implies that the combined ST258 OmpK35 and OmpK36 configuration would be advantageous in the face of Carbapenem exposure. We next aimed to investigate if the ST258 OmpK35 and OmpK36 configuration is advantageous, neutral or costly during infection. For this, we developed an acute murine infection model, in which KP is inoculated directly to the trachea and lung parenchyma via placement of an oral endotracheal tube. This mimics the inoculation route in ventilator-associated pneumonia (VAP) encountered in hospital settings[21].

We validated the accuracy and reproducibility of intubation and inoculum delivery using both standard microbiological techniques and in vivo imaging. First, we harvested the lungs immediately after inoculation and enumerated colony forming units (CFU) in the lung parenchyma (Supplementary Fig. 8a). The number of KP reaching the site of infection is reproducible across animals and accurately reflected the inoculation dose (e.g. 1000 CFU). Furthermore, we generated a bioluminescent tagged strain of ICC8001, where the *Photorhabdus luminescens* bacterial luciferase operon was introduced downstream of the *glmS* gene[22] (see details in the Method section). Following intubation of the tagged strain the total flux from the lung region was measured, confirming accurate administration across animals (Supplementary Fig. 8b, c). 3D-diffuse light imaging of this bioluminescent source immediately following inoculation demonstrates distribution deep into the dependent lung zones reproducing the common nidus of infection in the human host (Fig. 4a).

Once the accuracy of the model was confirmed we assessed whether OmpK35$_{ST528}$ and OmpK36$_{ST528}$ affect fitness in vivo. First, we inoculated 500 CFU (±10%) of ICC8001, a dose that results in a primary pneumonic focus and secondary bacteraemia. Following intubation, the mice developed a septic phenotype displaying severe dyspnoea, weight loss and become unresponsive to external stimuli. The infected mice reached the severity endpoint at 36 h post infection, when quantification revealed $2.00 \times 10^9$ CFU (mean, $n = 10$) in the lung tissue and $9.33 \times 10^4$ CFU/ml (mean, $n = 10$) in the blood.

We next compared the outcome of infection with each of the three isogenic strains (ICC8002-ICC8004) and with OmpK35$_{ST258}$/ΔOmpK36 (that functionally represents the ΔOmpK35/ΔOmpK36) to confirm the reported phenotype of the double porin mutant in our model[11] (Fig. 4b, c). Inoculating these strains individually illustrated that the OmpK35$_{ST258}$ truncation (ICC8003) or isoform variation in OmpK36$_{ST258}$ (ICC8002) alone or in combination (ICC8004) resulted in significant expansion in the lungs and dissemination to the blood. No significant differences from ICC8001 infection were observed in the total CFU from the lungs or dissemination to the blood. In contrast, the loss of both porins in OmpK35$_{ST258}$/ΔOmpK36 caused significant attenuation and this strain failed to reach a high pulmonary burden (mean $3.93 \times 10^5$, $n = 10$) associated with a low level of bacteraemia detectable in only one mouse ($n = 10$).

In order to more stringently determine if OmpK35$_{ST258}$ and OmpK36$_{ST258}$ have a fitness cost, we tested ICC8002, ICC8003 and ICC8004 in competition with ICC8001 utilising a fluorescent based in vivo competition assay. We chromosomally tagged the strains at the 3′ *glmS* site with either green (sfGFP) or red fluorescent (mRFP1) protein coding genes and confirmed

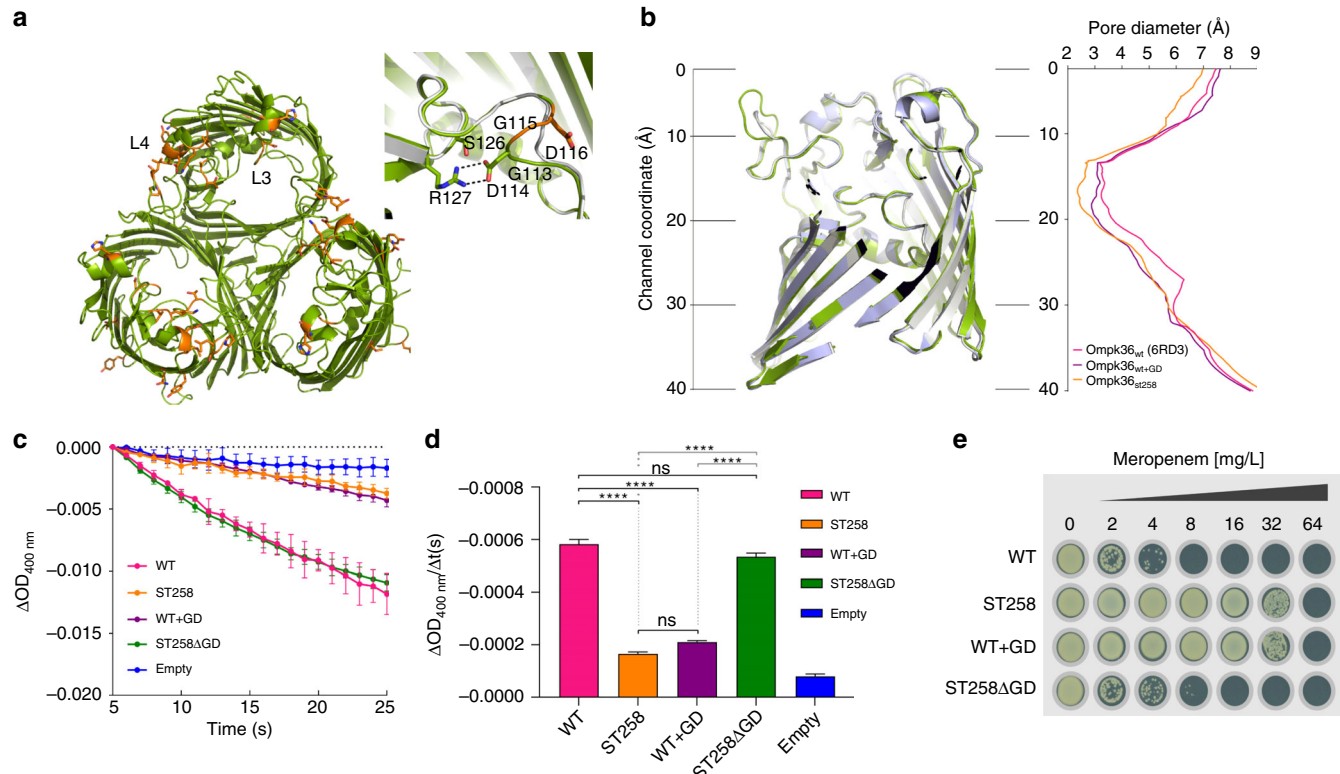

**Fig. 3** Gly-Asp insertion in L3 of OmpK36 reduces pore diameter, restricts Meropenem diffusion and mediates Meropenem resistance. **a** Cartoon representation of the OmpK36$_{ST258}$ trimer. The OmpK36$_{ST258}$ mutations have been mapped onto the structure. All mutations are shown as orange sticks. The majority of the mutations are found in L3/4. Inset: close-up view of the pore constriction zone. In order to accommodate the GD insertion, L3 has undergone a conformational change, stabilised by a salt-bridge, with subsequent constriction of the pore relative to the OmpK36$_{WT}$ (shown in grey cartoon). **b** Lateral view of OmpK36 monomer aligned to minimal pore diameter graph, calculated using the HOLE algorithm, demonstrating a reduction in minimal pore diameter in both OmpK36$_{ST258}$ and OmpK36$_{WT+GD}$ compared with OmpK36$_{WT}$. **c** OmpK36 isoforms were reconstituted into proteoliposomes and Meropenem diffusion was assessed by liposomal swelling assay. Meropenem influx is reduced in OmpK36$_{ST258}$ compared with OmpK36$_{WT}$. This effect is abolished by GD deletion (OmpK36$_{ST258\Delta GD}$) and reproduced by GD insertion (OmpK36$_{WT+GD}$) ($n = 3$ repeats, error bars = ± s.d.). **d** Calculated Meropenem uptake rate over 20 seconds ($\Delta OD_{400nm}/\Delta t(s)$) from (**c**). ****$p < 0.0001$, error bars ± s.e.m. **e** GD insertion in OmpK36 isoforms mediates Meropenem resistance assayed by agar dilution in KP (OmpK35$_{ST258}$ background with KPC-2). Substitution of OmpK36$_{WT}$ with OmpK36$_{WT+GD}$ increases the MIC to that of OmpK36$_{ST258}$, a phenotypic pattern reversed by GD deletion in the OmpK36$_{ST258\Delta GD}$ mutant

expression by fluorescent microscopy (Fig. S9a). We confirmed that the cost of fluorescent protein expression was silent by growth in vitro (Supplementary Fig. 9b, c). Furthermore, we verified neither tag confounded competition by co-inoculating ICC8001GFP and ICC8001RFP (250 CFU (±10%) of each), resulting in a 95% confidence interval traversing 50% (mean 51.3%, 95% confidence interval 47–55.7% in the lung and mean 47.12%, 95% confidence interval 37.4–56.8% in the blood) (Fig. 4d and S9). This demonstrated that in vivo competition could be used to assess relative fitness of ICC8002-ICC8004. In contrast to the results of single strain inoculations (Fig. 4b, c), when assessed in competition with ICC8001, ICC8003 (OmpK35$_{ST258}$) and ICC8002 (OmpK36$_{ST258}$) showed a trend towards reduced fitness in the lung (Fig. 4e, f), which reached significance in dissemination to the blood (Supplementary Fig. 9e, f). Importantly, ICC8001 completely out competed ICC8004 (expressing the dual ST258 combination of OmpK35/OmpK36 mutations), with no recoverable ICC8004 in either the lung (Fig. 4g) or blood (Fig. 4h).

In order to exclude the possibility that the out-competition of ICC8004 was due to pleiotropic effects of porin constriction on production of capsule, which is a key KP virulence factor[23], we quantified abundance of capsule between the different strains. This revealed no difference between ICC8001, ICC8002, ICC8003 and ICC8004 (Supplementary Fig. 10), suggesting that in the

absence of antibiotics the ST258 porin composition has a significant fitness cost in KP. Finally, we tested if the GD insertion alone, which mediates increased Carbapenem resistance, is sufficient to reproduce the observed virulence disadvantage during in vivo competition. For this, we carried out a series of infections using L3 GD mutants (on an OmpK35$_{ST258}$ background), competing OmpK36$_{WT}$ against OmpK36$_{WT+GD}$ and OmpK36$_{ST258}$ against OmpK36$_{ST258\Delta GD}$. In both sets of isogenic pairs the presence of the GD insertion resulted in almost complete out-competition in the severe murine pneumonia model (Fig. 4i). These data provide direct evidence that the structural determinant conferring porin constriction and retarding antibiotic entry disadvantages KP during infection.

The rapid dissemination of a single global resistant sequence type has occurred in other pathogens, such as ST131 *E. coli*, another frequently hospital-acquired pathogen. However, the traits conferring a resistant phenotype are proposed to be neutral, i.e. there is no attributable fitness cost in vivo[24]. The attenuation seen in the OmpK35$_{ST258}$/ΔOmpK36 was evident during infection, resulting in marked failure to replicate efficiently in the lungs and disseminate to the blood. Our OmpK36$_{ST258}$ variant occupies a middle ground; it is able to expand in vivo to high levels when strains containing this isoform are inoculated alone. However, it demonstrated a marked disadvantage in vivo when in competition with its WT counterpart. We infer from

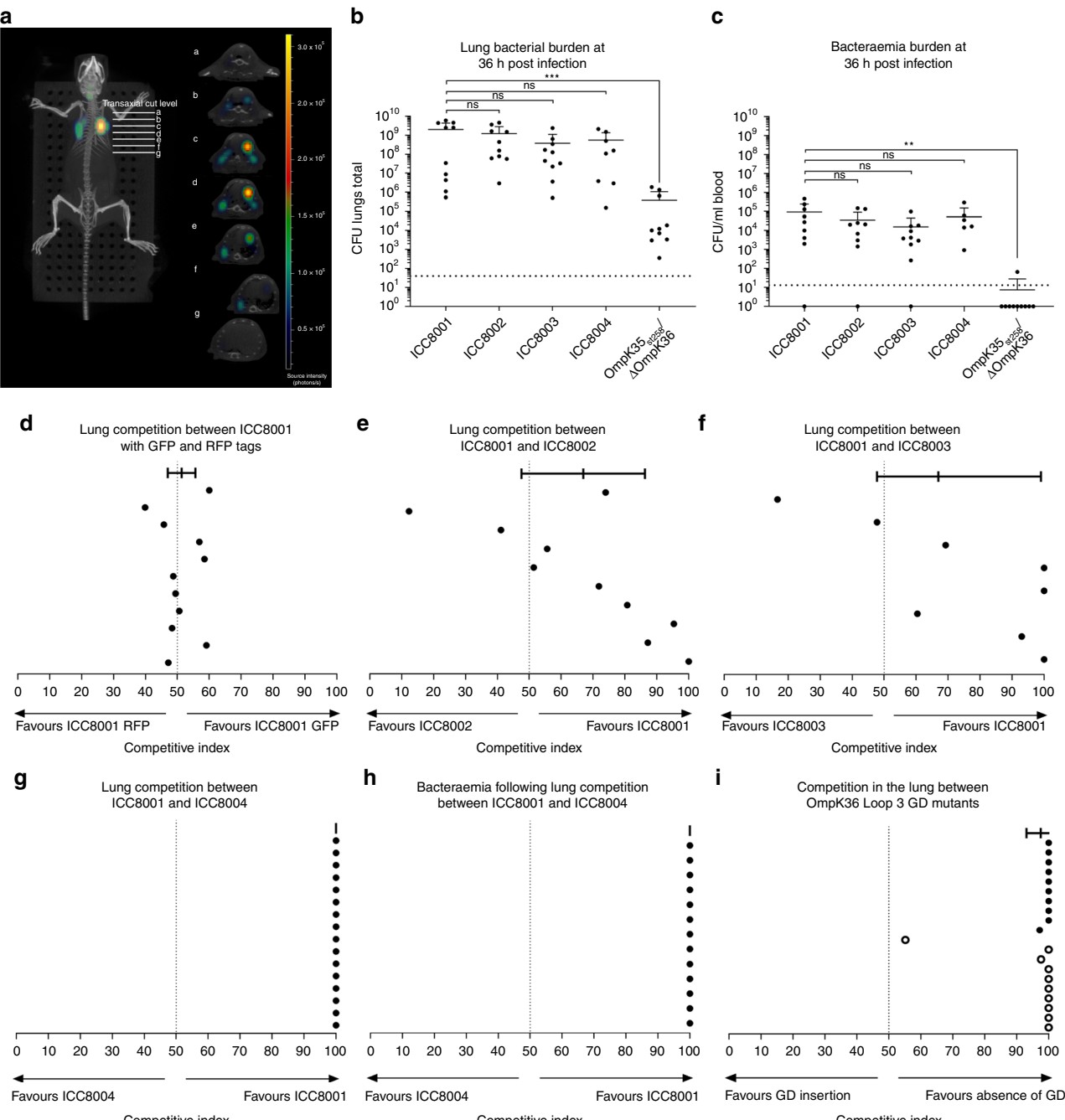

**Fig. 4** The Gly-Asp insertion in Asp L3 causes a fitness disadvantage in an in vivo model of Ventilator-associated pneumonia. **a** Delivery of bioluminescent KP to the lung parenchyma by intubation recorded by 3D-diffuse light imaging immediately post inoculation. Images were reconstructed into 3D using 3D living image. **b**, **c** Mice were intubated with 500 CFU of ICC8001-ICC8004 or OmpK35$_{ST258}$/ΔOmpK36 ($n = 10$/strain, error bars ± s.d.). Enumeration of CFU in lungs and blood, collected at 36 h post infection, revealed no significant difference between ICC8001 and all strains tested except OmpK35$_{ST258}$/ΔOmpK36. **$p < 0.001$, ***$p < 0.0002$. **d** Competition between ICC8001-GFP and ICC8001-RFP was tested by intubating 250 CFU or each competing strain. Enumeration of CFU of each strain in lungs and blood, identified by their colony fluorescence, was performed at 36 h post infection. Top bar represents mean value, points below represent individual mice ($n = 10$ per competition, error bars ± 95% confidence interval). No significant difference between ICC8001-GFP and ICC8001-RFP was detected in either the lungs or blood (Fig. S9d). **e** Competition between ICC8001 and ICC8002 (OmpK35$_{ST258}$) and **f** ICC8001 and ICC8003 (OmpK36$_{ST258}$) result in non-significant fitness disadvantage in the lung, but significant attenuation in dissemination to the blood (Fig S9e, f) ($n = 10$ per competition, error bars ± 95% confidence interval). **g**, **h** ST258 configuration at both OmpK35 and OmpK36 loci (ICC8004) results in complete out-competition by WT configuration (ICC8001) at 36 h post inoculation in the lung and dissemination to the blood, with no recoverable ICC8004 at either site ($n = 16$, error bars ± 95% confidence interval). **i** Competition between OmpK36 isogenic pairs with or without Gly-Asp Loop 3 insertion demonstrates that in vivo disadvantage results from pore constriction. Top bar represents mean value ($n = 20$ mice, error bars ± 95% confidence interval), OmpK36$_{WT}$ in competition with OmpK36$_{WT+GD}$ (open circles, each point represents 1 mouse ($n = 10$)) or OmpK36$_{ST258}$ in competition with OmpK36$_{ST258ΔGD}$ (closed circles, each point represents 1 mouse ($n = 10$))

these data, that in OmpK36$_{ST258}$, pore diameter reduction is a trade-off between Carbapenem resistance and retaining partial WT functionality. Indeed, we show that diffusion of lactose is reduced in the presence of L3 GD insertion. Whilst this is probably not a substrate utilised by KP in the murine host, it provides a putative mechanism leading to fitness cost in vivo; i.e. the resistance mutations may impair the ability of the strain to compete for resources that are limited during infection.

We provide compelling evidence that the molecular mechanism conferring resistance is actually disadvantageous in vivo. Given KP ST258 success, we cannot rule out the existence of fitness costs in other resistance mutations in successful clades, the elucidation of which could help in our continual fight against the spread of multidrug-resistant pathogens.

Clinically, our data points towards a model where selection pressure imposed by utilisation of Carbapenems in hospital settings, drives the expansion of ST258 KP. Accordingly, we provide evidentiary support that we should continue to Start Smart-Then Focus[25] in order to reduce the selective pressure in healthcare, and implement restrictive prescribing policies aimed at minimising the inappropriate use of broad-spectrum agents such as Carbapenems.

## Methods

**Generation of isogenic strains and tagging.** Strains (S), plasmids (V, vector) and primers (P) used in this study are listed in the supplementary material (Table 1, S1, S2 and S3, respectively). All genomic deletions and substitutions were made in ICC8001 (strain 1 (S1) in Table S2) using a two-step recombination methodology resulting in scarless and markerless mutants as previously described[26]. The isogenic strains generated are summarised in Table 1. Unless otherwise stated, all vectors were generated by Gibson Assembly (New England Biolabs).

We generated the OmpK35$_{ST258}$ truncation coding sequence (FJ577672) in vitro. The OmpK35$_{WT}$ open reading frame (ORF) was amplified from ICC8001 with primers P1/2 and assembled into linearised V1 (P3/4) generating V3. The frame-shift mutation was introduced by site-directed mutagenesis (P5/6) generating V4.

All OmpK36 isoforms were introduced into ΔOmpK36 mutant of ICC8001. The OmpK36$_{WT}$ ORF deletion vector (V5) was generated by assembling 500 bp upstream and downstream flanking regions of the gene (P9/10 and P11/12), into linearised pSEVA612S (P13/14).

OmpK36$_{ST258}$ was amplified (P15/16) and inserted between these flanking regions to generate a substitution vector (V6). OmpK36$_{ST258ΔGD}$ (V7) was made by site-directed mutagenesis of V6 using primers (P17/18). To generate OmpK36$_{WT+GD}$ we first amplified the WT ORF (P19/20) into the deletion vector (V5) to generate V8. We inserted G115D116 coding sequence on V8 by PCR (P21/22).

We generated a vector (V9) to insert the *P. luminesens* luciferase operon (*luxCDABE*) at the Tn7 insertion site downstream of the *glmS* locus. Upstream (P23/24) and downstream (P25/26) 500 bp regions were amplified by PCR, ligated together followed by ligation into digested pSEVA612S (V1) with KpnI followed by BamHI and RcoRI, respectively. *luxCDABE* was amplified from V10 by PCR (P27/28) and ligated between the up and downstream homology regions using Nco1 and Xho1 sites generating V9.

We replaced the *luxCDABE* cassette in V9 by inverse PCR (P29/30) of this vector and replaced it with sfGFP or mRFP1 amplified from V11 (P31/32) and V12 (P31/32). These vectors express sfGFP1 (V13) and mRFP1 (V14) from a constitutive promoter after genomic integration at the Tn7 site downstream of *glmS*. This was screened by PCR (P33/34) external to the locus and confirmed by amplicon sequencing and fluorescence. OmpK36$_{ST258ΔLSP}$ and OmpK36$_{ST258R127A}$ were generated by site-directed mutagenesis with primers 35/36 and 37/38, respectively.

**Growth curves.** Saturated overnight cultures in LB were diluted 1:100 in 200 µl of fresh LB or M9 media supplemented with glucose or lactose at 0.4% w/v. OD$_{600nm}$ readings were taken on a FLUOStar Omega (BMG Labtech, UK) at 30 min intervals with shaking at 200 rpm at 37 °C between readings.

**Transfer of KPC-2 and OXA-48 carbapenemases.** Transfer of resistance plasmids was achieved by conjugation into KP using an *E. coli* plasmid transformant donor. Plasmids were extracted from KPST258 (KPC-2 containing plasmid, S4) and KPOXA-48 (OXA-48 containing plasmid, S5) using a Bacterial Artificial Chromosome miniprep kit (Zymo Scientific, CA, USA). Room temperature DH5a (S6) competent cells[27] were transformed by electroporation. Transformants were confirmed by PCR for the presence of resistance genes (KPC-2 P39/40, OXA-48 P41/42).

Saturated overnight cultures of DH5a transformant donors and ICC8001 (and derived lines) receivers containing pACBSR (SmR) were combined 8:1 (v/v). The mixture was diluted 1:40 in PBS and 40 µl incubated on an LB plate at 37 °C for 6 h. The transconjugants were plated on LB Agar supplemented with streptomycin (50 mcg/ml), ertapenem 0.5 or 1 mcg/ml. KPC-2 and OXA-48 presence in transconjugants was confirmed by PCR and pACBSR was cured by serial passage in LB containing only ertapenem at an appropriate concentration to maintain the carbapenemase resistance plasmid.

**Preparation of capsule and of outer membrane proteins.** Capsule production in the isogenic KP strains ICC8001-4 was measured through quantification of capsular uronic acid as described[28]. Isogenic strains were grown to an OD$_{600}$ of 0.8–1 in either LB or M9 media and the outer membrane porin proteins purified as previously described[29] with the alteration of the sonication amplitude to 38%. SDS-PAGE electrophoresis analysis was performed using 12% acrylamide Mini-protean TGX precast gels.

**Minimum inhibitory concentrations.** Broth MICs were conducted by the reference laboratories at Public Health England (Colindale, UK) and resistance defined by EUCAST breakpoints[30].

Plates for agar dilution MICs were made in accordance with published guidance[31] using Mueller-Hinton Agar (Merck, UK) supplemented with meropenem, ertapenem or imipenem. Saturated overnight cultures of test strains were diluted in 0.8% saline and 20 µl (representing $10^4$–$10^5$ CFU) plated and incubated overnight at 37 °C.

**Overexpression and purification of OMPs.** The synthetic genes for OmpK36$_{wt}$ and OmpK36$_{ST258}$ (GeneArt, Thermofisher) were designed with a 6xHis-tag and a tobacco etch virus (TEV) cleavage site between the outer membrane targeting sequence and OmpK36 mature domain. The synthetic genes were subcloned into pEBMSCHIS (V16), and were transformed in porin deficient *E. coli* BL21(DE3) Omp8 (S7) competent cells (*Δlamb ompF::Tn5 ΔompA ΔompC*)[32]. Outer membranes were prepared as before[33] and purified in 10 mM HEPES pH 7, 150 mM NaCl and 0.4% $C_8E_4$.

**Liposome swelling assays.** Proteoliposomes were prepared as previously described with no modifications[34]. The change in OD$_{400}$ of the mixture was measured for 90 s (1 s reading intervals) with a SpectraMax M Series Multi-Mode Microplate Readers (Molecular Devices). A 5 s reading delay was imposed for all measurements in order to reduce initial reading spikes. The isotonic concentration of each substrate was empirically determined by measuring the change in OD$_{400}$ of control liposomes; isotonic solutions showed less than >0.001 units change over 90 s. The first 20 s of the measurements were used for analysis and plotting since they represent the linear decrease of OD$_{400}$. Each curve represents three separate liposome reconstitutions.

**Crystallisation.** OmpK36$_{WT}$, OmpK36$_{ST258}$ and OmpK36$_{WT+GD}$ were exchanged into 10 mM HEPES pH 7, 150 mM LiCl, and 0.4% $C_8E_4$ using a PD-10 desalting column (GE Healthcare) and concentrated to 10 mg/ml. Plate-like crystals for OmpK36$_{ST258}$ were grown from a solution containing 0.1 M NaCl, 0.1 M LiSO$_4$, 0.1 mM Tris HCl pH 8.5 and 30% PEG400 at 4 °C. Needle-like crystals for OmpK36$_{WT}$ and OmpK36$_{WT+GD}$ were grown from a solution containing 0.1 M Lithium sulphate, 0.1 M sodium citrate pH 5.6 and 12% PEG4000 at 20 °C. The OmpK36$_{ST258}$ crystals were cryoprotected by supplementing the crystallisation condition with 25% ethylene glycol and were frozen in liquid nitrogen. OmpK36$_{WT}$ and OmpK36$_{WT+GD}$ crystals were directly frozen in liquid nitrogen. Diffraction screening and data collection were performed at Diamond Light Source synchrotron.

**Data collection.** OmpK36$_{ST258}$ data to 3.23 Å were collected on I04 at Diamond Light Source and processed using autoPROC and STARANISO[35]. The space group was determined to be $P2_12_12$ with 18 copies of OmpK36$_{ST258}$ in the asymmetric unit. OmpK36$_{WT+GD}$ data to 2 Å were collected on I24 at Diamond Light Source and processed using xia[36]. The space group was determined to be *C2* with three copies of OmpK36$_{WT+GD}$ in the asymmetric unit. OmpK36$_{WT}$ data to 1.92 Å were collected on I04 at Diamond Light Source and processed using xia[36]. The space group was determined to be *P21* with six copies of OmpK36$_{WT+GD}$ in the asymmetric unit. Data collection statistics are summarised in Table 2.

**Structure solution and refinement.** All structures were determined by molecular replacement in Phaser[37] using the OmpK36$_{Q235R}$ structure (PDB ID: 5O79)[13] as search model. Initial refinement of OmpK36$_{ST258}$ to 3.23 Å was carried out in REFMAC5[38] and at later stages in Phenix[39] with non-crystallographic symmetry (NCS). After rigid body and restrained refinement extra electron density corresponding to the mutations and insertions were identified, built and refined. Additional density, possibly detergent molecules, that was observed on the surface of the protein was not modelled due to the low resolution. The OmpK36$_{WT}$ and OmpK36$_{WT+GD}$ structures were refined to 1.92 and 2.0 Å, respectively, in Phenix[39]

with NCS. All model building was performed in Coot[40]. Refinement statistics are summarised in Table 2.

**Pore size analysis**. Constriction of the pore diameter was determined by measurement of the channel path by HOLE[41].

**Animal studies**. Animal studies were performed on project license 70/8413 granted by the United Kingdom Home Office under the Animals (Scientific Procedures) Act 1986; ethical approval was granted by the Imperial College Animal Welfare and Ethical Review Body. All animal work complied with relevant ethical regulations for animal testing and research and results are reported in line with the Animal Research: Reporting of In Vivo Experiments (ARRIVE) guidelines (http://www.nc3Rs.org.uk/arrive-guidelines).

Female BALB/c mice (8–10 week 18–20 g) (Charles River, UK) were housed under a 12 h light/12 h dark light cycle with access to food and water ad libitum. Anaesthesia was induced by i.p. administration of ketamine (80 mg/kg) and medetomidine (1 mg/kg), and procedural recovery took place at 35 °C following the administration of atipamezole (1 mg/kg) reversal. Intubation was achieved by placing a 21 G plastic cannula, over a fibre-optic light illuminated cable, through the glottis under direct vision, of mice suspended by their incisors according the kit protocol (Kent Scientific, CT, USA). In single strain infections inoculum was generated by the dilution of saturated overnight culture (LB) in 1xPBS (final volume 50 μl, final dose 500 CFU). This was pipetted into the hub of the cannula and inoculum drawn into the lungs by the inspiratory effort of spontaneously breathing animals. Two 400 μl (approximate to tidal volume) air flushes were used to evenly distribute the inoculum to the distal airways and CFU counts confirmed in the inoculum by plating onto LB Agar and overnight incubation at 37 °C.

3D bioluminescent imaging for the distribution post intubation was carried out using the IVIS SpectrumCT (Caliper Life Sciences, Massachusetts, USA). Inoculum (ICC8001LUX) was prepared by sub-culture of saturated overnight 1:100 in 5 ml LB and grown at 37 degrees at 200 rpm. Sub-culture was concentrated (10×) by centrifugation and resuspension in LB. Fifty microlitres was delivered as outlined above. Mice were immediately imaged by microCT scan and acquisition of bioluminescence at three emission wavelengths (500 nm/520 nm/540 nm). Images were reconstructed in Living Image 4.5.5 as per the manufactures guidance (Caliper Life Sciences, Massachusetts, USA).

For time course experiments mice were housed in groups of 5. At 36 h post infection blood was collected ante-mortem by venepuncture of the lateral tail. Lungs were collected post-mortem by open dissection of the thorax and tissue homogenised in 3 ml of sterile PBS in a tissue homogeniser (Miltenyi Biotec). Serial dilutions of blood and lung homogenate were plated out in triplicate on LB Agar (with 100 mcg/ml Rifampicin) and colonies counted following overnight incubation at 37 °C. Each strain was tested in a total of 10 animals, 5 animals per group in biological duplicate.

For competition assays, overnight cultures were mixed in a 1:1 ratio before dilution in PBS to a total dose of 500 CFU, composed of 250 CFU of each tagged competitor. In each competition assay the tags identifying competing strains were reversed in a biological replicate providing a total number of 10 mice for each competition tested. Inoculum was delivered using the same methodology as for single inoculum infections. Inoculum was enumerated by overnight incubation on LB Agar plates and fluorescent tags identified by blue light transillumination (Safe Imager, ThermoFisher Scientific) of colonies. In order to calculate competition ratios serial dilutions of lung homogenate and blood were plated out in triplicate and plates were transilluminated after overnight incubation at 37 °C. The number of recovered colonies expressing sfGFP or mRFP1 was used to calculate the competition ratio, representing as a percentage of the total CFU. In all assays we present the data relative to the number of ICC8001 (WT) recovered, i.e. 100% represents 100% of colonies counted being ICC8001 and none of the competing strain or 50% representing half recovered colonies being ICC8001 and half the competing strain.

**Statistics**. Data was analysed in Prism (Graphpad Software, La Jolla, California, USA). Error bars are described in legends depicting either SD or SEM. Rates for meropenem influx were calculated over the first 20 s of data recording (5–25 s) by non-linear regression without imposed constraints. A multiple comparisons ANOVA, with replacement of $n$ for replicates with the degree of freedom from the non-linear regression was used to calculate significance by Tukey's multiple comparisons.

CFU counts in the lung and the blood and capsule uronic acid quantification were compared between groups using a non-parametric ANOVA (Kruskal-Wallis test) followed by Dunn's multiple comparisons test against the mean rank of ICC8001 (WT) as a control. 95% confidence interviews were calculated in Prism (Graphpad Software, La Jolla, California, USA).

Non-significance represents any $p > 0.05$, otherwise $p$ values are presented in the legend of figures.

**Reporting summary**. Further information on research design is available in the Nature Research Reporting Summary linked to this article.

## Data availability

The source data underlying Figs. 1c–e and 3b–d, 4b–i, Supplementary Fig. 4a, b, c, e, Supplementary Fig. 5b–f, Supplementary Fig. 7a–c and Supplementary Fig. 8a–c are available in the Source Data File. The coordinates and structure factors of PDB 6RD3 (OmpK36$_{WT}$), 6RCP (OmpK36$_{ST258}$) and 6RCK (OmpK36$_{WT+GD}$) are available in the Protein Databank.

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

## Acknowledgements

All animal work described in this study was funded by grants from the Medical Research Council and the Wellcome Trust. We would like to thank Diamond Light Source for beam time allocation and access, and Dr Marc Morgan, Imperial College London, for help with scheduling. S.J.B. acknowledges support from the NIHR Imperial Biomedical Research Centre (BRC). K.B. is supported by a grant for the Medical Research Council (MR/N020103/1). J.L.C.W. is supported by a clinical training programme from the MRC (MRC CMBI Studentship award MR/R502376/1). G.F. is supported by an Investigator grant from the Wellcome Trust.

## Author contributions

J.L.C.W. designed the overall strategy and performed the molecular biology, biochemistry and in vivo experiments; he also analysed the data and wrote the paper. L.E.K. performed biochemical and in vivo experiments, she analysed data and wrote the paper. M.R. and H.S.K. performed the structural and proteoliposome assays. They analysed data. W.W.L. analysed the capsule. SJB provided clinical input and edited the paper. A.C. participated in supervision, data analysis and edited the paper. K.B. led the structural and proteoliposome aspects of the project, participated in data analysis and wrote the paper. G.F. led the in vivo aspects of the project, participated in data analysis and wrote the paper.

## Additional information

**Competing interests:** The authors declare no competing interests.

