## [Peer Review File · Nature Communications]

Reviewers' Comments:

Reviewer #1:

Remarks to the Author:

Summary

In this work the authors show that carbapenem resistance in the clinical *Klebsiella pneumoniae* (KP) ST258 strain is mediated by the presence of plasmid encoded carbapenemase genes such as KPC-2 and OXA-48 and the loss/truncation of OmpK35 and modification of OmpK36. Their studies also show that modification of these Omp proteins confers a significant fitness cost in vivo (reduced ability to replicate in lungs and disseminate to the blood compared to WT). The presence of a GD insertion in loop 3 OmpK36ST258 not seen in OmpK36WT results in a conformational change in loop 3 allowing it to extend into the pore. This is further stabilized by the formation of an Asp-Arg dyad and contributes to a reduction of the OmpK36ST258 pore diameter. This pore constriction is thought to confer carbapenem resistance. Carbapenem diffusion studies show a reduction in antibiotic diffusion when proteins were reconstituted in liposomes. In addition, the authors show that carbapenem resistance in ST258 KP requires both porin modification and the expression of carbapenemase enzymes. In the absence of either KPC-2 or OXA-48 the strains tested remained sensitive to carbapenems. Apart from a few critiques and corrections that should be addressed by the authors, this is a strong body of work.

Supplementary Figure 2

Though antibiotic diffusion is reduced in OmpK36ST258 we see an increase in fitness cost probably due to restricting the influx of nutrients etc. These porins are non-selective and allow transport of small molecules, as is seen in the experiments with glucose. Glucose is a small molecule with a molecular weight ~ 180 g/mol while the carbapenem ertapenem is 475.5 g/mol. Why wasn't a molecule with a molecular weight comparable to that of ertapenem used as a control to show how pore restriction affects diffusion? Moreover, this might also give insight into how a decrease in nutrient influx may affect fitness cost. Classical liposome swelling assays would give a better description of nutrient size limitations.

Correct caption. Change D to F in second line from the bottom.

Supplementary Table 5

- Fix alignment in column 3, 3rd line from the bottom
- The values in the highest resolution shell for I/σ of OmpK36WT+GD and OmpK36WT and the CC1/2 of OmpK36WT are low. Adjust the resolution cutoff.
- Authors should be consistent with significant figures reported for resolution in the text as well as in the refinement statistics table.

Results and Discussion

Figure 1C. Why is the protein band for OmpK35 not seen in the WT strain ICC8001 while OmpK36 is seen?

Table 1B. The isogenic strains remain sensitive to ampicillin and so shouldn't AMP be colored green and not red as shown in the table? Please check the colors.

Page 7, line 138: are the authors referring to Table 1B? There is no Table 2.

Page 7, line 138: Avibactam (Table 2B). What are the authors referring to?

Page 9, line 191: the authors discount the effects of the L4 insertions on pore reduction because the conformational change seen was away from the constriction zone. Were studies done with an OmpK36WT+LSP construct? Or did the authors just assume that because this conformational change was away from the constriction zone it did not play a role in conferring resistance? These studies should be added.

The Asp-Arg salt bridge seems to be important in stabilizing loop 3. Do the authors have any thoughts on whether disrupting this Asp-Arg dyad by altering pH for example would affect the protein structure in the OmpK36ST258 or OmpK36WT+GD? Does charge play a role in nutrient uptake, and is this changed in the mutant?

At what pH were these crystals obtained?

Grammar

- Page 5 line 97 of Introduction: correct font size of "usage"
- Table S1, caption A: correct the spelling of carbapenem
- Methods, page 1, last paragraph: correct font size

Reviewer #2:

Remarks to the Author:

Wong et al present a study examining the role of outer membrane porin mutations in conferring carbapenem resistance in *Klebsiella pneumoniae* clinical isolates. The authors determine the effect of the porins in mediating clinical levels of drug resistance through analysis of isogenic strains. Examining the structure and diffusion activities of purified porin isoforms, the authors show that the variant conferring drug resistance in clinical isolates restricts the influx of carbapenem antibiotics due to a constricted pore. Sensitive tests of competitive fitness in a mouse infection model reveal fitness costs of the pore-constricted variant that lie in a middle ground between WT and complete loss of the porin. The study reveals the mechanism behind the resistance of a clinically successful and highly prevalent porin mutant. The study was carefully performed and conclusions were justified, and the findings will be of broad interest to the field.

The manuscript can be improved by addressing details outlined here:

Table S2 is insufficient given the number of *Klebsiella* strains tested in this study, and should list each strain described in the text (for example, of the isogenic strains only ICC8001 is listed, but the table should list ICC8002, 3, etc and other engineered strains, along with the respective genotype, to aid in analysis of the paper).

Table and Figure numbering should be checked throughout the manuscript as several were mislabeled. For instance, line 138 and 140 should say Table 1B, line 226 should say Fig. 3A, line 464 should say Fig. S4.

Line 402, in Table 1 legend the acronyms should be checked for accuracy. CZA is listed as referring to ceftazidime -- please verify. "CAZ" is used in the table but is not defined in the legend.

line 448, Fig. 3A legend should indicate exact time post-infection that the signal was measured.

Line 261: how was significance determined when analyzing difference in fitness between mutant and parent bacteria in blood?

Based on the structure the authors determine that the loop3 insertion is stabilized by a salt-bridge between two residues which were present in the WT porin -- has the role of the salt bridge in conferring resistance/carbapenem restriction been examined?

NCOMMS-19-11944A

Response to reviewers' comments:

We would like to thank both reviewers for their very positive assessment and constructive comments. Below are point-by-point responses to their reports.

Reviewer #1

Comment 1: Supplementary Figure 2. Though antibiotic diffusion is reduced in OmpK36ST258 we see an increase in fitness cost probably due to restricting the influx of nutrients etc. These porins are non-selective and allow transport of small molecules, as is seen in the experiments with glucose. Glucose is a small molecule with a molecular weight ~180 g/mol while the carbapenem ertapenem is 475.5 g/mol. Why wasn't a molecule with a molecular weight comparable to that of ertapenem used as a control to show how pore restriction affects diffusion? Moreover, this might also give insight into how a decrease in nutrient influx may affect fitness cost. Classical liposome swelling assays would give a better description of nutrient size limitations.

Response: We have addressed this point with additional experiments using two further carbohydrates in swelling assays - stachyose (666g/mol) and lactose (342g/mol). While we observe no diffusion of stachyose, lactose diffuses through OmpK36_{WT} but not through OmpK36_{WT+GD} (Figure S3E). Consistently, growth of strains expressing OmpK36_{WT+GD} is impaired in a medium containing lactose as the only carbon source (Figure S3F).

Comment 2: Correct caption. Change D to F in second line from the bottom.

Response: Corrected

Comment 3: Supplementary Table 5 - Fix alignment in column 3, 3rd line from the bottom

Response: Now Supplementary Table 6, alignment corrected.

Comment 4: The values in the highest resolution shell for I/σ of OmpK36_{WT+GD} and OmpK36_{WT} and the CC1/2 of OmpK36_{WT} are low. Adjust the resolution cutoff.

Response: The reported low values of I/σ and CC1/2 for the OmpK36_{WT+GD} were a typo during the completion of the table. This has now been corrected. The OmpK36_{WT} suffered by some anisotropy, but we agree with the reviewer that the values were low; anisotropic correction did not benefit this data set since the values improved but the completeness decreased. We decided to rescale the data at lower resolution, 1.98Å. The structure was re-refined and re-deposited to the PDB (same accession number). The validation report has also been uploaded to the submission.

Comment 5: Authors should be consistent with significant figures reported for resolution in the text as well as in the refinement statistics table.

Response: We have corrected the significant figures throughout.

Comment 6: Figure 1C. Why is the protein band for OmpK35 not seen in the WT strain ICC8001 while OmpK36 is seen?

Response: OmpK35 is seen in both ICC8001 and ICC8002 (arrow). This has now been made clearer.

Comment 7: Table 1B. The isogenic strains remain sensitive to ampicillin and so shouldn't AMP be colored green and not red as shown in the table? Please check the colors.

Response: ICC8001 and the derived isogenic strains are all resistant to AMP (chromosomal-mediated); accordingly it is coloured red.

Comment 7: Page 7, line 138: are the authors referring to Table 1B? There is no Table 2.

Response: Yes, this has been corrected.

Comment 8: Page 7, line 138: Avibactam (Table 2B). What are the authors referring to?

Response: This has been corrected in antibiotic key.

Comment 9: Page 9, line 191: the authors discount the effects of the L4 insertions on pore reduction because the conformational change seen was away from the constriction zone. Were studies done with an OmpK36WT+LSP construct? Or did the authors just assume that because this conformational change was away from the constriction zone it did not play a role in conferring resistance? These studies should be added.

Response: We have now experimentally confirmed this conclusion by showing that deletion of the LSP from OmpK36 did not affect the MIC (new Figure S4).

Comment 10: The Asp-Arg salt bridge seems to be important in stabilizing loop 3. Do the authors have any thoughts on whether disrupting this Asp-Arg dyad by altering pH for example would affect the protein structure in the OmpK36ST258 or OmpK36WT+GD? Does charge play a role in nutrient uptake, and is this changed in the mutant?

Response: We thank the reviewer for this comment. We have now tested the effect of an R127A substitution on uptake and MIC. This has shown that the salt bridge is not required for Carbapenem resistance and that the GD insertion alone is sufficient to constrict the pore and hinder diffusion (Figure S5); this is now described in the main text (lines 211 to 217).

Comment 11: At what pH were these crystals obtained?

Response: As reported in the Methods section, OmpK36_{ST258} was crystallised at pH 8.5 and OmOmpK36_{WT} and OmpK36_{WT+GD} at pH 5.6.

Comment 12: Grammar

Response: We thanks the reviewer for spotting the errors, which have now been corrected.

Reviewer #2

Comment 1: Table S2 is insufficient given the number of Klebsiella strains tested in this study, and should list each strain described in the text (for example, of the isogenic strains only ICC8001 is listed, but the table should list ICC8002, 3, etc and other engineered strains, along with the respective genotype, to aid in analysis of the paper).

Response: We thank the reviewer or this comment. Details of all strains used in this study are now presented in a new Table (Table S3).

Comment 2: Table and Figure numbering should be checked throughout the manuscript as several were mislabeled. For instance, line 138 and 140 should say Table 1B, line 226 should say Fig. 3A, line 464 should say Fig. S4.

Response: Again, we thank the reviewer or this comment. These errors have now been corrected.

Comment 3: Line 402, in Table 1 legend the acronyms should be checked for accuracy. CZA is listed as referring to ceftazidime -- please verify. "CAZ" is used in the table but is not defined in the legend.

Response: This has been corrected.

Comment 4: line 448, Fig. 3A legend should indicate exact time post-infection that the signal was measured.

Response: This has been added to the legend.

Comment 5: Line 261: how was significance determined when analyzing difference in fitness between mutant and parent bacteria in blood?

Response: We calculated 95% confidence intervals around the mean (n=10 individual mice for each condition tested). Graphically this is plotted as the mean with 95% confidence interval error bars (as described in the legend). In addition, we provide the reader with competition ratio from each animal tested below the summary point. The inference of this method is that, according to our experimental data, we are 95% confident that the true competition value lies within these ranges. If this range traverses 50% then there is no advantage (or disadvantage) to either strain tested in the assay. The chance of the true value lying outside the 95% confidence interval range is 5%. This is equivalent to the accepted $p < 0.05$ (i.e. 1 in 20 chance of 5%) significance cut-off. In addition, when ICC8001 is competed against ICC8004 no ICC8004 was recovered, resulting in 100% competition (100% favouring ICC8001 and 0% favouring ICC8004) with a 95% confidence interval of 100% to 100%. In these experiments, we are therefore 100% confident that the true result is 100%.

Comment 6: Based on the structure the authors determine that the loop3 insertion is stabilized by a salt-bridge between two residues which were present in the WT porin -- has the role of the salt bridge in conferring resistance/carbapenem restriction been examined?

Response: We thank the reviewer for this comment. Please see our response to reviewer 1 comment 10.

Reviewers' Comments:

Reviewer #1:

Remarks to the Author:

The authors have addressed all of my comments satisfactorily.

Reviewer #2:

Remarks to the Author:

The authors have addressed all of my concerns with this resubmitted manuscript. The authors provide additional work that explains how channel alterations driving resistance can modulate fitness, with tests of carbohydrate diffusion and growth in defined growth medium (lactose). The new results with defined carbohydrates are emphasized in the abstract and main text but the data are shown in the supplemental files. The manuscript overall has been improved.